# Identification of Mobile Colistin Resistance Gene *mcr-10* in Disinfectant and Antibiotic Resistant *Escherichia coli* from Disinfected Tableware

**DOI:** 10.3390/antibiotics11070883

**Published:** 2022-07-01

**Authors:** Senlin Zhang, Honghu Sun, Guangjie Lao, Zhiwei Zhou, Zhuochong Liu, Jiong Cai, Qun Sun

**Affiliations:** 1College of Biomass Science and Engineering, Sichuan University, Chengdu 610064, China; zslforest@gmail.com; 2Irradiation Preservation Key Laboratory of Sichuan Province, Chengdu Institute of Food Inspection, Chengdu 611135, China; sunhonghu901@hotmail.com (H.S.); caijiongcj@gmail.com (J.C.); 3Key Laboratory of Bio-Resource and Eco-Environment of the Ministry of Education, College of Life Sciences, Sichuan University, Chengdu 610064, China; laoguangjie@stu.scu.edu.cn (G.L.); zhouzhiwei@stu.scu.edu.cn (Z.Z.); liuzhuochong_hmc@stu.scu.edu.cn (Z.L.)

**Keywords:** disinfectant resistance, colistin, multidrug resistance, *mcr-10*, plasmid, *E. coli*

## Abstract

The widespread escalation of bacterial resistance threatens the safety of the food chain. To investigate the resistance characteristics of *E**. coli* strains isolated from disinfected tableware against both disinfectants and antibiotics, 311 disinfected tableware samples, including 54 chopsticks, 32 dinner plates, 61 bowls, 11 cups, and three spoons were collected in Chengdu, Sichuan Province, China to screen for disinfectant- (benzalkonium chloride and cetylpyridinium chloride) and tigecycline-resistant isolates, which were then subjected to antimicrobial susceptibility testing and whole genome sequencing (WGS). The coliform-positive detection rate was 51.8% (161/311) and among 161 coliform-positive samples, eight *E. coli* strains were multidrug-resistant to benzalkonium chloride, cetylpyridinium chloride, ampicillin, and tigecycline. Notably, a recently described mobile colistin resistance gene *mcr-10* present on the novel IncFIB-type plasmid of *E. coli* EC2641 screened was able to successfully transform the resistance. Global phylogenetic analysis revealed *E. coli* EC2641 clustered together with two clinically disinfectant- and colistin-multidrug-resistant *E. coli* strains from the US. This is the first report of *mcr-10*-bearing *E. coli* detected in disinfected tableware, suggesting that continuous monitoring of resistance genes in the catering industry is essential to understand and respond to the transmission of antibiotic resistance genes from the environment and food to humans and clinics.

## 1. Introduction

The development of antimicrobial resistance (AMR) has led to difficulties in the clinical treatment of serious bacterial infections. In response to global efforts to reduce AMR by minimizing antibiotic use, biocides (e.g., disinfectant and preservative) have increasingly been used as an important component of infection control [1], especially during the COVID-19 pandemic [2]. These biocides, including quaternary ammonium disinfectant, aldehydes, alcohols, phenols, bisphenols, and halogenic compounds, are currently considered the first line of defense against foodborne pathogens in hospitals or food processing facilities due to the versatility and efficiency of their chemical active ingredients [3]. Among them, quaternary ammonium disinfectant is non-corrosive and non-irritating, with low toxicity and high antimicrobial efficacy over a wide pH range, making them a widely used surface disinfectant and cleaning agent in food processing and production environments [4]. Benzalkonium chloride (BAC) and cetylpyridinium chloride (CPC), the most commonly used members of quaternary ammonium disinfectant, have broad-spectrum (i.e., bacteria, algae, fungi, and viruses) antimicrobial activity [5]. As a result, they are widely used as a surface disinfectant in food processing lines (e.g., poultry facilities, cleaning, and sanitizing facilities), dairy/agricultural environments, health care facilities, and home oral care [6,7].

However, this raises questions about the possible role of quaternary ammonium disinfectants in promoting the development of antimicrobial resistance, particularly multidrug resistance to antimicrobials. The epidemiological relationship between antibiotic resistance and higher MIC values of quaternary ammonium disinfectant in clinical *E. coli* isolates has been confirmed [8]. Recent studies have shown that BAC exposure can induce antibiotic resistance through multiple genetic mechanisms, including the coexistence of BAC and antibiotic resistance genes on the same mobile DNA molecule, mutations in the *pmrB* (colistin resistance) gene, and the overexpression of efflux pump genes [9]. Russell et al. mentioned that the frequent use of CPC could likewise result in bacterial resistance [10]. The use of quaternary ammonium disinfectant in food production and processing environments may not be as effective as expected, which provides selection pressure for strains with acquired resistance to other antimicrobial agents [11].

Resistance to most antimicrobials and a lack of novel antimicrobials against Gram-negative bacteria can lead to the reuse of older antibiotics. For example, colistin (polymyxin E), a cationic polypeptide antibiotic, was one of the first antibiotics to have a significant effect against Gram-negative bacteria [12]. It binds to the negatively charged lipopolysaccharide (LPS) of the outer membrane of Gram-negative bacteria, leading to membrane rupture and ultimately to cell death. Colistin and tigecycline are considered to be the remedy and last line of defense for serious bacterial infections [13,14]. Unfortunately, mobile colistin resistance genes (*mcr-1* to *mcr-10*) and tigecycline resistance genes (*tetX* to *tetX6*) carried by plasmids have recently been found to be horizontally transmitted with plasmids; the emergence of these genes accelerates the fall of the last line of defense against antibiotics [15,16]. Of the ten *mcr* gene variants, *mcr-1* is common in more than forty countries on six continents [17]. *mcr-10* is a novel allele, first isolated in ascites from a patient in China in 2020, and has been identified in animals, humans, and the environment [16,18,19]. However, little attention has been paid to *mcr-10* resistant isolates from disinfected tableware environments. Potentially, this is an important route of *mcr-10* transmission from the environment and food to healthy humans.

Previous studies on the multidrug resistance of bacterial disinfectants and antibiotics have mainly focused on hospital clinical infectious bacteria and rarely on multidrug resistance of foodborne bacteria to disinfectant and antibiotic. Additionally, studies on foodborne bacteria in tableware have mostly focused on the evaluation of disinfection effects, and few studies have been conducted on disinfectant- and antibiotic-resistant strains. Meanwhile, *E. coli*, as an important opportunistic pathogen among foodborne bacteria, is one of the common outbreak factors of foodborne diseases in China. Therefore, monitoring multidrug resistance to disinfectant and antibiotic in *E. coli* from disinfected tableware in the catering industry is important to understand the spread of resistance genes in foodborne bacteria and to regulate sanitization practices. This study aimed to monitor resistant *E. coli* to commonly used disinfectants and the last antibiotic in disinfected tableware to understand their resistance characteristics.

## 2. Results

### 2.1. Strain Identification and Antimicrobial Susceptibility

In our study, a total of 161 coliform-positive disinfected tableware samples were detected from 311 tableware samples, with a detection rate of 51.8%. From the 161 coliform-positive disinfected tableware samples, eight isolates of quaternary ammonium disinfectant and tigecycline resistance were detected. They were identified as *E. coli* by API-20E biochemical identification and whole genome sequencing. For all eight isolates, resistance to tigecycline, benzalkonium chloride, and cetylpyridinium chloride was detected, with the MIC values recorded of 2, 16, and 8 µg/mL, respectively. The majority (6/8) exhibited resistance to multiple antimicrobial agents other than meropenem (Table 1). Ampicillin resistance (*n* = 8) and tetracycline resistance (*n* = 6) were commonly detected in 6/8 quaternary ammonium disinfectant resistant isolates, followed by cefoxitin (*n* = 2), ceftriaxone sodium (*n* = 2), and chloramphenicol (*n* = 2). In addition, resistance phenotypes to colistin and polymyxin B were observed in isolates EC2639 and EC2641. All eight *E. coli* strains exhibited multidrug resistance characteristics, with TGC-AMP-TET-BAC-CPC being the most frequently observed resistance profile. Multilocus sequence typing (MLST) showed that eight strains were identified in seven ST types: ST2795, ST1571, ST4537, ST218, ST3907, ST3076, and ST5783.

### 2.2. ARG Characterization of Multidrug-Resistant Positive E. coli

In silico analysis showed that all strains contained the *mdf* (A) gene, a multidrug efflux pump gene with broad spectrum specificity that mediates antibiotic resistance to erythromycin, tetracycline, rifampin, kanamycin, chloramphenicol and ciprofloxacin, in addition to resistance to quaternary ammonium disinfectants [20]. Additionally, most (7/8) isolates carried *formA*, a gene resistant to formaldehyde (preservative) (Table 2) [21]. The quinolone resistance gene *qnrS1*, β-lactam resistance-associated genes *bla*_TEM-1A_, *bla*_ZEG-1_, *bla*_OXA-10_, and *bla*_MIR-2_, third-generation cephalosporins resistance-related gene *bla*_CTX-M-55_, aminoglycoside resistance-related genes *strB* and *aadA1*, tetracycline resistance gene *tet* (A), trimethoprim resistance-related gene *dfrA14* and sulfonamide resistance-related gene *sul2* were detected in 2 (25.0%), 8 (100.0%), 1 (12.5%), 2 (25.0%), 2 (25.0%), 2 (25.0%), and 1 (12.5%) of these isolates, respectively. It was worth noting that the recently described mobile colistin resistance gene *mcr-10* was identified in the isolate EC2641. No mobile tigecycline resistance genes (*tet* variants) were found. The plasmid replicon results showed that all strains had 1–5 plasmid replicon types, with IncFIB-type plasmids being the most common type, followed by IncY, IncFIA, and Col440I.

### 2.3. Genetic Characterization of Colistin-Resistant E. coli EC2641

Long-read whole genome sequencing of *E. coli* EC2641 yielded 1.93 clean gigabytes, second generation short read length whole genome sequencing yielded 1.204 GB of clean sequence, mixed assembly of second- and third-generation sequencing data, screening to one complete chromosome sequence and two complete plasmid sequences. The sizes were 4.74 Mb, 129.9 kb, and 198.3 kb, with G + C contents of 50.9, 51, and 48.55%, respectively. Among them, the mobile colistin resistance gene *mcr-10* was located on the lncFIB (K)-type plasmid of size 123.9 kb and no other resistance genes were found on this plasmid. In addition, the acquired resistance gene *mdf* (A), *bla*_ZEG-1_, and *formA* were found in the chromosomal DNA of this strain. The MLST typing database and serotype database were queried and the strain EC2641 was identified as ST3907 and O112ab:H9. Virulence factor predictions indicated that the isolate encodes a uropathogenic specific protein USP and is an *E. coli* causing urinary tract infections.

### 2.4. Conjugation of mcr-10 among E. coli Isolate EC2641 under Laboratory Conditions

Conjugation experiments showed that the colistin resistance gene *mcr-10* was able to transfer into the recipient bacterium J53 with a conjugation frequency of 5.03 × 10^−4^. The *mcr-10* gene could be detected by PCR (Appendix A) in both the parental strains and their transconjugants. In the case of the transconjugants, colistin MIC values were re-evaluated and found to be a 4-fold (0.5 µg/mL to 2 µg/mL) increase compared to the recipient bacterium J53.

To understand the origin of the plasmid carrying *mcr-10* in *E. coli* EC2641, a sequence comparison of plasmid pMCR_10_2641 was performed on NCBI. The results of the five plasmids with the highest similarity were characterized (Appendix A). The most closely related plasmid to pMCR_10_2641 was the unnamed plasmid from *E. coli* strain A1_180 (GenBank Accession Number NZ_CP040383.1), with 45.34% coverage and up to 99.99% nucleotide identity. *E. coli* strain A1_180 was isolated from seagull feces from mudflats in Anchorage, the USA in 2016. A comparison of the pMCR_10_2641 plasmid sequence with the top five plasmid sequences of similarity revealed that the *xerC*-*mcr-10* structure is unique to pMCR_10_2641 (Figure 1). A genomic island (GI) of 8.7 kb in size was identified at 4.5 kb upstream of *mcr-10*, which had eight open reading frames (ORF) encoding the type 3 fimbriae of bacterium.

No genes associated with the bacterial type IV secretion system (T4SS) and relaxase gene mediating conjugative transfer were identified in the plasmid pMCR_10_2641. However, the plasmid had two 285 bp origin of transfer sites (*oriT*). Another plasmid with a size 198.3 Kb in this strain, p2641_2, was found to carry a locus encoding a protein related to the bacterial type IV secretion system. Combined with the results of the conjugative experiments (Appendix A), plasmid pMCR_10_2641 was shown to be a novel non-self-transmissible mobile plasmid that may mediate the mobilization of the colistin resistance gene *mcr-10* by the helper plasmid p2641_2.

### 2.5. Genetic Characterization of mcr-10 Carrying Plasmids

To understand the horizontal transfer mechanism of *mcr-10*, the genetic environment of *mcr-10* was characterized by local covariance analysis (Figure 2). The six *mcr-10*-bearing plasmid sequences, including pMCR_10_2641, were isolated in this study and the remaining five plasmids were from animals (*n* = 1), hospital wastewater (*n* = 2), and clinical samples (*n* = 2). Among them, pMCR10_090065 is the first isolated and reported to harbor *mcr-10* gene. Compared to the recently described genetic environment of *mcr-10*, our *mcr-10* was linked to the upstream *xerC* gene, except that IS*Ec36* was interrupted by the insertion sequence IS*kos1Δ* in plasmid pMCR10_090065, and the downstream formed a conserved *xerC*-*mcr-10*-IS*Ec36* structure with the insertion sequence IS*Ec36*.

To further understand the origin of *mcr-10* in *E. coli* non-self-mobilizable plasmids, a comparative analysis of plasmid sequences carrying *mcr-10* reported by NCBI and articles was performed (Figure 3). A total of 20 complete plasmids sequences carrying *mcr-10* were retrieved on NCBI (Appendix A), including *Enterobacter roggenkampii* (30%, 6/20), *Raoultella ornithinolytica* (10%, 2/20), *Enterobacter cloacae* (15%,3/20), *Cronobacter sakazakii* (5%, 1/20), *Citrobacter freundii* (5%, 1/20), *Enterobacter hormaechei* (5%, 1/20), *Enterobacter kobei* (5%, 1/20), *Enterobacter asburiae* (5%, 1/20), *Enterobacter* sp. (10%, 2/20), and *Klebsiella quasipneumoniae* (10%, 2/20), which demonstrated its wide host range. Typing of these plasmids revealed a smaller range of host plasmids for *mcr-10*, concentrated in three plasmid replicon types, lncFIB (*n* = 12), lncFIA (*n* = 2), and lncFII (*n* = 4). Structural comparisons of plasmids carrying *mcr-10* revealed the insertion sequence IS*903* upstream of *xerC-mcr-10*, which is immediately downstream of the mobile element IS*Ec36* (except for individual plasmids where this position is interrupted).

### 2.6. Upregulation of mcr-10 Expression at Subinhibitory Concentrations of Colistin

To understand whether *mcr-10* expression is inducible in the original strain, four drug stresses were selected to investigate the expression characteristics of *mcr-10* (Appendix A). The relative expression of this gene was significantly upregulated 2-fold in 2 µg/mL (1/2 MIC) colistin (*p* < 0.001), and no significant changes were observed after the disinfectants benzalkonium chloride (1/2 MIC, 8 µg/mL) and cetylpyridinium chloride (1/2 MIC, 4 µg/mL) disinfectant stresses. *mcr-10* was upregulated 1.5-fold after IPTG induction. Two disinfectants, benzalkonium chloride and cetylpyridinium chloride, did not affect the expression of the mobile colistin resistance gene *mcr-10*. The results of the induction assay showed that the *mcr-10* gene could be specifically induced by colistin in *E. coli* EC2641. It was shown that the original host of *mcr-10*, *E. coli* EC2641 can adapt rapidly under colistin stress.

### 2.7. Global Phylogenetic Analysis of E. coli Carrying the mcr Variant

Core SNP-based phylogenetic analysis was performed to determine the epidemiological relevance of the isolate EC2641 to publicly available *E. coli* isolates (*n* = 208) that have the *mcr* gene in the NCBI database (Appendix A). Isolate EC2641 was clustered with two clinical isolates (ST32) from the United States carrying the disinfectant resistance gene *qacE*. Our isolate EC2641 was clustered into a large evolutionary branch including humans (clinical), animals, the environment, and food (*n* = 141) from various countries (USA, Germany, Czech Republic, Egypt, Canada, Australia, Vietnam, and China) that exhibit global epidemiological characteristics (Figure 4).

## 3. Discussion

We used the resistance profile of *E. coli* to quaternary ammonium disinfectants and the last antibiotic to indirectly assess the risk of potential resistance transmission of coliform bacteria in disinfected tableware. Our data suggest that the detection of coliform in disinfected tableware is of concern, which may involve the latent presence of multiple pathogenic bacteria. Consistent with previous studies on disinfectant-resistant *E. coli* isolated from hospitals [8], the *E. coli* we isolated also exhibited multidrug resistance. The identification of multidrug-resistant *E. coli* to tigecycline and colistin in our study suggests that this may pose risks and challenges for our last resort antibiotics. Although no mobile tigecycline resistance genes (*tet* variants) were found in our study, these eight *E. coli* strains still exhibited tigecycline resistance, which poses a concern for last resort antibiotic therapy. Bacterial efflux pumps were the main mechanism of tigecycline resistance before it was found that mobile tigecycline resistance genes were located in plasmid-mediated bacterial resistance. For example, previous studies have shown that mutations in the *adeS* gene can lead to the overexpression of the AdeABC efflux pump, resulting in the decreased susceptibility of *A. baumannii* to tigecycline [24,25,26]. Antibiotics and disinfectants have different bactericidal effects on bacteria and are used in different settings, but bacteria have a similar response strategy to disinfectants and antibiotics, namely the bacterial efflux pump mechanism [10]. Additionally, the mechanism of tigecycline resistance in *E. coli* isolated in this study requires further experimental confirmation.

Most disinfectant resistance genes are carried by chromosomes that are usually part of the bacterial efflux pump system, including the resistance-nodulation-division (RND) superfamily, the main promoter superfamily (MFS), the multidrug and toxic compound extrusion (MATE) family, small multidrug resistance (SMR) family, and the ATP binding cassette (ABC) super family [27]. A previous study on disinfectant resistance in *E. coli* isolated from retail meat in Sichuan, China, found that the most frequent disinfectant resistance genes were *ydgE/ydgF*, *mdf* (A), and *sugE* located on chromosomes [11]. Zhou et al. identified the chromosomally encoded quaternary ammonium disinfectant resistance gene *mdf* (A) exhibiting the highest prevalence in *E. coli* isolated from retail meat in the United States [28]. Few disinfectant resistance genes located in mobile elements were found. This is consistent with the detection of the quaternary ammonium disinfectant resistance gene *mdf* (A) in our *E. coli* (100%, 8/8). Multilocus sequence typing (MLST) showed that *mdf* (A) was identified in seven STs, suggesting that its presence in *E. coli* is widespread and may provide a genetic advantage for the spread of AMR bacteria.

Since the first report of the colistin resistance gene *mcr-1* in 2015 [29], *mcr* variants have been widely identified in various species of different origins worldwide. Nine *mcr* homologs (*mcr-2* to *mcr-10*) have been identified [18]. Plasmid-mediated *mcr* genes have spread worldwide and pose a high threat to public health networks. Currently, the novel colistin resistance gene variant *mcr-10* gene has been recently monitored in animals, healthy humans (clinical), hospital wastewater, and raw milk [16,19,22,23]. Genetic structural characterization of the antimicrobial resistance plasmid pMCR_10_2641 indicated that the plasmid we identified carrying the antimicrobial resistance gene *mcr-10* is a probable novel plasmid. The resistance gene *mcr-10* is adjacent to a virulence island carrying locus encoding the type 3 fimbriae system of *Klebsiella pneumoniae* (*mrkABCDFJ*), allowing the host to attach to surfaces and form biofilms to resist the action of disinfectants and antibiotics, which are associated with virulence in *Enterobacteriaceae* [30,31]. Importantly, we report for the first time that a strain of *E. coli* carrying *mrc-10* is multidrug-resistant to disinfectants and antibiotics isolated from disinfected tableware in Chengdu, China, which not only expands the source range of *mcr-10* but also suggests that the spread of the novel colistin resistance gene *mcr-10* is implicated in humans.

It is generally believed that nonconjugative plasmids cannot be transferred horizontally to the recipient bacterium, yet this ignores the role of other plasmids carrying binding modules generated in the original strain. Xu et al. demonstrated that *Klebsiella pneumoniae* can transfer nonconjugative virulence plasmids (containing *oriT* sites) and conjugative plasmids under conditions that reduce extracellular polysaccharides or use *E. coli* as an intermediate strain [32]. Consistent with previous studies, the *mcr-10*-bearing plasmid we identified lacked a binding module and is not self-transmissible, but our experimental results suggest that *mcr-10* may be transferred horizontally into the recipient bacteria via a type IV secretion system carried by the helper plasmid. This helper plasmid employs the *oriT* of the mobilized plasmid to accomplish mobilization, which is consistent with a previous study [33].

Unlike the plasmid preference of *mcr*-*1* (IncI2, IncX4) [34], *mcr-10* prefers lncFIB-type plasmids. The *mcr-10*-bearing IncFIB-type plasmid pMCR_10_2641 is an antimicrobial resistant and virulence co-existing plasmid, which may have important implications for clinical treatment. In fact, IncFIB-type plasmids usually carry multiple resistance genes, and a recent study reported a novel resistance gene cluster *tnfxB1*-*tmexCD1*-*toprJ1* located in an IncFIB-type plasmid carrying multiple resistance genes (*strAB*, *armA*, *aph(3’)- ia*, *qnrB4*, *sul1*, *mphA*, *mphE*, *msrE*, and *bla*_dha1_) [35]. Despite the structural diversity of *mcr-10*-bearing plasmids, the genetic environment of *mcr-10* is generally the same, mostly composed of elements such as *xerC*, IS*26*, IS*903*, and IS*Ec36*. Unusually, the plasmid pMCR_10_2641 in this study is inserted immediately after the sequence IS*Ec36* with a serine site-specific recombinase encoding gene *pinR*. *xerC* and *pinR* are tyrosine site-specific recombinant genes and serine site-specific recombinant genes, respectively, both of which are integrons. *xerC*-type tyrosine recombinase can mediate the transfer of its surrounding carbapenem resistance gene *bla*_NMC_ through site-specific recombination [36], but it is unclear whether the integrin *pinR* has a mobilizing effect on *mcr-10*. The covariance results suggest that *mcr-10* is in the midst of a complex insertion sequence with mobile elements and that the *xerC-mcr-10*-IS *Ec36* structure may be an important structure leading to the horizontal transfer of *mcr-10*. The genetic environment of *mcr-10* in all six plasmids highlights the importance of the insertion element IS*Ec36* in this genetic structure, and the results combined with plasmid comparisons suggest that the insertion sequences IS*903* and IS*Ec36* may be closely associated with *xerC*-mediated specific recombination. The conserved genetic environment and transferability between different bacterial hosts may lead to the widespread availability of *mcr-10* containing colistin-resistant isolates from different sources.

In fact, little is known about whether the transcription of resistance genes can be affected by disinfectant exposure and thus lead to antibiotic resistance [37,38]. In the present study, we confirmed no significant change in *mcr-10* expression under disinfectant stress, suggesting that *mcr-10* transcription may not be affected under disinfectant stress. Furthermore, induction with IPTG did not result in significant changes in *mcr-10* expression, which is consistent with previous studies in which the MIC of colistin remained unchanged [18]. The expression of *mcr-10* was significantly upregulated 2-fold in *E. coli* EC2641 under colistin stress, which was similar to the observations of Xu et al. in *E. roggenkampii* [19], and also indicates that *mcr-10* is functional for the host to cope with the selective pressure of colistin. Disinfectant exposure plays an important role in promoting the development of antibiotic resistance. It has been shown that bactericides used for disinfection can enhance antibiotic resistance in Gram-negative bacteria; for example, *Burkholderia lata* developed resistance to ceftazidime, imipenem, and ciprofloxacin after exposure to low concentrations of benzalkonium chloride and was found to be associated with significant upregulation of outer membrane proteins and ABC transporter proteins [39]. In cells adapted to benzalkonium chloride, new resistance was most frequently found to ampicillin (eight species), cefotaxime (six species), and sulfamethoxazole (three species), some of them with relevance for healthcare-associated infections such as *Enterobacter cloacae* or *E. coli* [40]. In *A. baumannii* ATCC17978, BAC increased the MIC of several aminoglycoside antibiotics (kanamycin, tobramycin, streptomycin, gentamicin, and amikacin) [41].

Global phylogenetic analysis showed that *mcr*-bearing *E. coli* exhibited a global epidemic profile (USA, Germany, Czech Republic, Egypt, Canada, Australia, Egypt, Vietnam, and China, etc.) and our isolate EC2641 was clustered with 141 *E. coli* from different sources including humans, the environmental, food and animals, supporting the direct/indirect transmission between humans and animals and the environment, which is consistent with the recent global concept of spreading mobilized colistin resistance genes from the environment to humans [16,42,43]. It is worth noting that *E. coli* EC2641 showed a close evolutionary relationship with two *E. coli* (ST 32) strains from the USA clinically carrying the *mcr-9* and quaternary ammonium disinfectant resistance gene *qacE*. In early studies, *E. coli* ST32 was isolated from the feces and hides of cattle in the United States [44], indicating that ST32 is not host-specific and is readily transmitted from animals to humans. We reasonably speculate that disinfectant-resistant bacteria may be important vectors of globally important antimicrobial resistance gene mobilization. The use of disinfectants in environmental disinfection has exploded since the COVID-19 epidemic, and monitoring of the multidrug resistance of such disinfectants with other antibiotics (e.g., tigecycline, colistin, imipenem, and meropenem) warrants further study. We emphasize the importance of disinfectants in the development and spread of bacterial antibiotic resistance.

## 4. Materials and Methods

### 4.1. Bacterial Isolation and Species Identification

This study was carried out during the period between June 2019 and December 2020. A total of 311 disinfected tableware samples were collected from 119 restaurants (using chemical disinfectant) in Chengdu, China. The fast test paper for coliform of tableware (NANJING SAN-AI, Nanjing, China) wet with sterile phosphate-buffered saline were used to sample the collected tableware samples, following the sampling principle of tableware surface in oral and food contact. Coliform detection sheets were placed in a 37 °C incubator for overnight culture. When the paper turned yellow or showed red spots on a yellow background, it was judged to be positive for coliform. Coliform were collected from positive paper with wet sterile cotton swabs for further analysis.

The wet swabs of coliform-positive samples were enriched in 30 mL of buffered peptone water (BPW, Oxoid, Hampshire, UK) and incubated overnight at 37 °C. Benzalkonium chloride, cetylpyridinium chloride, and tigecycline were used for the isolation of antibiotic and disinfectant resistant bacteria. Briefly, the overnight incubated culture was treated with benzalkonium chloride and cetylpyridinium chloride at a final concentration of 2000 µg/mL for 30 min at room temperature, and then screened for tigecycline resistant *E. coli* on the CHROMagar^TM^ Orientation plate (CHROMagar^TM^, Paris, France) containing 2 µg/mL tigecycline. Further strain identification was determined by API-20E biochemical testing and whole genome sequencing.

### 4.2. In Vitro Antimicrobial Susceptibility Testing

The broth dilution method was used for determining the minimum inhibitory concentration (MIC) of polymyxin B, colistin, tigecycline, ceftazidime, ceftriaxone, meropenem, ampicillin, chloramphenicol, cetylpyridinium chloride, and benzalkonium chloride. Resistance to polymyxin B, colistin, and tigecycline was determined according to the European Committee on Antimicrobial Resistance Testing (EUCAST). Clinical breakpoints (https://eucast.org/clinical_breakpoints/, accessed on 5 October 2021) and minimum inhibitory concentrations (MICs) for benzalkonium chloride and cetylpyridinium chloride MICs higher than those of standard strain *E. coli* ATCC 25922 were considered as resistant. Resistance to the other antibiotics was interpreted according to the CLSI instructions [45], with *E. coli* ATCC 25922 used as quality control strains.

### 4.3. Transconjugation Assay

Transconjugation assay was performed according to a previously described method [15]. The transferability of *mcr-10* in *E. coli* EC2641 was determined by the filtration membrane method. Briefly, *E. coli* J53 (resistant to sodium azide) was the recipient strain, and strain EC2641 (carrying *mcr-10*) was the donor strain in this study. Firstly, the donor and recipient were cultured in LB medium to logarithmic phase, and then the donor and recipient were mixed and inoculated on 0.22 filter membrane in the ratio of 1:3 and cultured overnight at 37 °C. Putative transconjugants were screened in LB agar medium containing 150 µg/mL sodium azide and 2 µg/mL colistin, and the successful transconjugants were identified by PCR amplification of the target gene (*mcr-10*) [18].

### 4.4. Induced Expression, RNA Extraction, Real-Time Reverse Transcription PCR (RT–PCR)

For induction assays, isolate EC2641 was grown in LB without antibiotic until OD600 = 0.5. Overnight culture of the bacteria was diluted in fresh LB broth (1:100, *v/v*) with final colistin concentration of 2 µg/mL, benzalkonium chloride concentration of 4 µg/mL, cetylpyridine chloride concentration of 4 µg/mL, and IPTG concentration of 1 mmol/L [18]. The antibiotic-free medium was used as the control. The culture was incubated at 37 °C for 10–12 h and harvested by centrifugation at 4 °C. RNA was extracted using a small amount of the total RNA extraction kit (TIANMO BIO, Beijing, China) according to the manufacturer’s instructions, followed by genomic DNA elimination and cDNA synthesis using the PrimeScript™ RT Reagent Kit with gDNA Eraser (Takara Bio USA, San Jose, CA, USA). gDNA removal was confirmed using PCR. RNA size, integrity and total amount were determined using a BioDrop µLite+ (Biochrom, Cambridge, UK).

The primers used for RT-qPCR were designed using Primer Premier 6.0 (Appendix A). The amplification efficiency of all primer pairs was tested using standard dilution procedures. RT-qPCR analysis was conducted on a QuantStudio™ 3 Real-Time system with SYBR green fluorescence dye. The *16s* gene was used as a reference control for normalization. The relative differences in gene expression were calculated as a fold change using the formula 2^−ΔΔCT^ [19].

### 4.5. DNA Extraction, Whole-Genome Sequencing

A single colony of *E. coli* isolates was selected and cultured in LB medium at 37 °C. Genomic DNA was extracted using a Gentra Puregene Yeat/Bact.Kit (Qiagen, Chatsworth, CA, USA). The harvested DNA was detected by the agarose gel electrophoresis and quantified by a Qubit^®^ 2.0 Fluorometer (Thermo Scientific, Waltham, MA, USA). According to the manufacturer’s instructions, the harvested DNA was subjected to WGS on the Illumina NovaSeq 6000 system (Illumina, San Diego, CA, USA), which generated 150-bp paired-end reads from a library with an average insert size of 350 bp. The *E. coli* EC2641 was further sequenced by a Nanopore PromethION platform (Nanopore, Oxford, UK) following a 10-Kbp library protocol.

### 4.6. Sequence Assembly, Annotation and Bioinformatic Analysis

For each isolate, >550 Mbp high-quality clean paired-end reads were obtained and de novo assembled using SPAdes v.3.9.0 [46]. In addition, the *E. coli* EC2641 hybrid assembly of short Illumina reads and long PromethION reads was performed using Unicycler v0.4.8 [47]. Then, we compared the reads to the assembled sequence, counted the distribution of sequencing depth, distinguished whether the assembled sequence was a chromosomal sequence or a plasmid sequence according to sequence length and alignment, and checked whether it is a circular genome.

The prokka version 1.14.6 with default parameters was used to call the blast+ to search for a small core set of well-characterized proteins and the Hidden Markov Model (HMM) to identify coding regions to accomplish rapid annotation of bacterial genomes and plasmid sequences [48]. Serotypes, sequence types, virulence genes, and antimicrobial resistance genes were identified using SerotypeFinder version 2.0.1, Multi-Locus Sequence Typing (MLST) version 2.0.4, VirulenceFinder version 2.0.3, and ResFinder version 4.0.1 from the Center for Genomic Epidemiology (CGE) [49]. Genes with 80% identity and greater than 90% coverage were considered present, and genes with coverage between 40% and 90% were considered present but partial genes. The IslandPath-DIOMB version 0.2 was used to predict GIs based on both the detection of dinucleotide bias in eight genes or more and the identification of a mobility gene in the same region [50].

The annotated GBK files were uploaded to the online tool oriTfinder to identify the classification of plasmids (conjugated, mobilizable, and non-mobilizable) by using the integration of the profile HMM-based relaxase gene search module with a BLAST-based *oriT* sequence search module [51]. The plasmid replicon genotype and insertion sequence (IS) elements of the plasmid were identified using PlasmidFinder version 2.0.1 and ISfinder [52,53], replicons and IS with 80% identity and greater than 90% coverage were considered present, and replicons with coverage between 40% and 90% were considered present but partial replicons. All plasmid sequences carrying *mcr-10* were searched on NCBI. BLAST Ring Image Generator (BRIG) and Easyfig v2.2.5 were used for comparative analysis of plasmids and generation of physical maps [54,55]. The above bioinformatics analysis tools used default parameters if not otherwise specified.

### 4.7. Phylogenetic Analysis

Phylogenetic analysis based on single nucleotide polymorphisms (SNPs) was performed for the *E. coli* isolate EC2641 carrying *mcr-10* gene. Our isolate EC2641 was compared to the publicly available genomes of *E. coli* in the NCBI database, updated on January 14th, 2022, that carried a variant of the *mcr* gene (*n* = 208). Isolates were mapped to the reference *E. coli* complete genome ATCC25922 (CP032085.1) for all the collected *mcr*-carrying isolates. Metadata for the selected *E. coli* sequences from NCBI were collected (Appendix A). SNPs variant calling were determined using Snippy v4.6.0 (https://github.com/tseemann/snippy, accessed on 15 January 2022) and the output files were combined into a core SNPs alignment using Snippy core. Maximum likelihood phylogenetic trees were then generated from SNPs alignment using RAxML-NG [56] and the trees were visualized with iTOL [57].

### 4.8. Accession Numbers

WGS data of all eight isolates, including two complete plasmid sequences of *E. coli* EC2641, were deposited in GenBank database (accession no. PRJNA818460).

### 4.9. Statistical Analysis

Statistical analysis was performed by the modular Student *t*-test using Prism 9 software (GraphPad Software, San Diego, CA, USA). Results are shown as the mean ± SD of the three individual test results. *p*-values < 0.05 were considered statistically significant.

## 5. Conclusions

In summary, we recovered eight strains of quaternary ammonium disinfectant- and antibiotic-resistant *E. coli* in disinfected tableware and characterized their resistance profile and the prevalence of the recently identified colistin resistance gene *mcr-10* in *E. coli* isolates. The non-self-transmissible novel plasmid carrying *mcr-10* identified in this study was mobilized by a helper plasmid, highlighting the potential widespread ability of *mcr-10* to the environment and humans. To achieve the “One Health” strategy for humans, the characterization and spread of bacterial resistance in the sanitized environment associated with the food industry should be continuously monitored. Therefore, monitoring the multidrug resistance profile of disinfectants and antibiotics in tableware can be an important reference for preventing and controlling the spread of resistance genes.

## Figures and Tables

**Figure 1 antibiotics-11-00883-f001:**
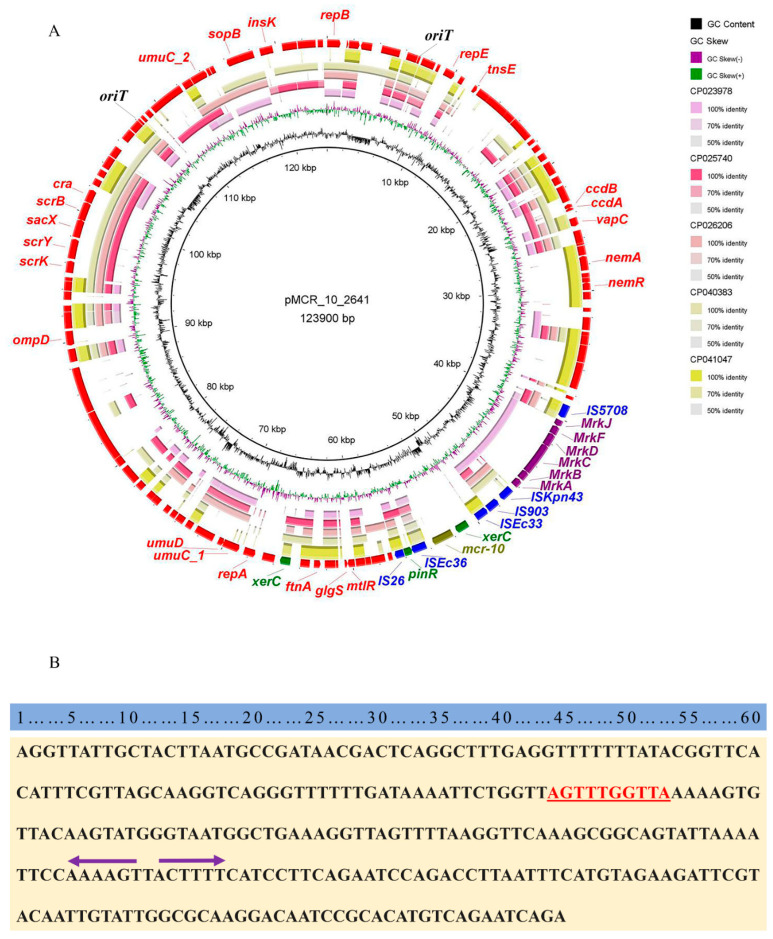
Genetic structural characterization of the antimicrobial resistance plasmid pMCR_10_2641. (**A**) Comparison of pMCR_10_2641 with the five highest matched plasmids in the NCBI reference database. The out-layer (red color) represents the plasmid pMCR_10_2641 in this study, which was used as the reference plasmid for a sequence comparison. Green arrows in the outer circle indicate integrons, blue arrows indicate various insertion sequences, and purple arrows indicate complete virulence islands. Colistin resistance gene *mcr-10* is highlighted by the olive. The arrow direction indicates the direction of transcription. The graph was generated by BRIG v0.95. (**B**) Schematic of the origin of the transfer sites (*oriT*) structure. The red font indicates the conserved nick region; The purple arrow is a pair of inverted repeats (IRs).

**Figure 2 antibiotics-11-00883-f002:**
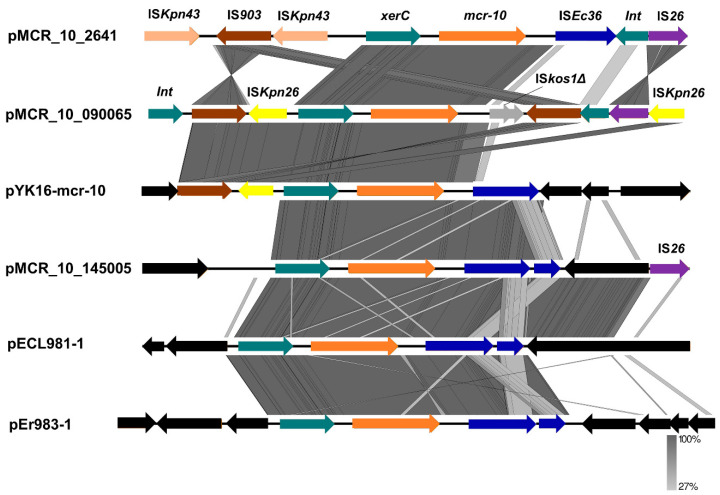
Comparison with the recently reported genetic environment of *mcr-10*. pMCR_10_2641 is the plasmid isolated for this study, pMCR_10_090065 the first plasmid identified and reported for *mcr-10* [18], pMCR_10_145005 plasmid was obtained from the stool samples of healthy volunteers in Chengdu [22], pYK-*mcr-10* is the most recent plasmid isolated from animal sources [23], pECL981-1 and pEr983-1 are the most recent plasmids isolated from hospital wastewater [19]. Orange arrows represent the drug resistance gene *mcr-10*, dark cyan arrows indicate site-specific recombinase-encoding genes (integrons), black arrows indicate other functional genes, and the remaining colored arrows represent the corresponding insertion sequences or mobile progenitors. The graph was generated by easyfig v2.2.5.

**Figure 3 antibiotics-11-00883-f003:**
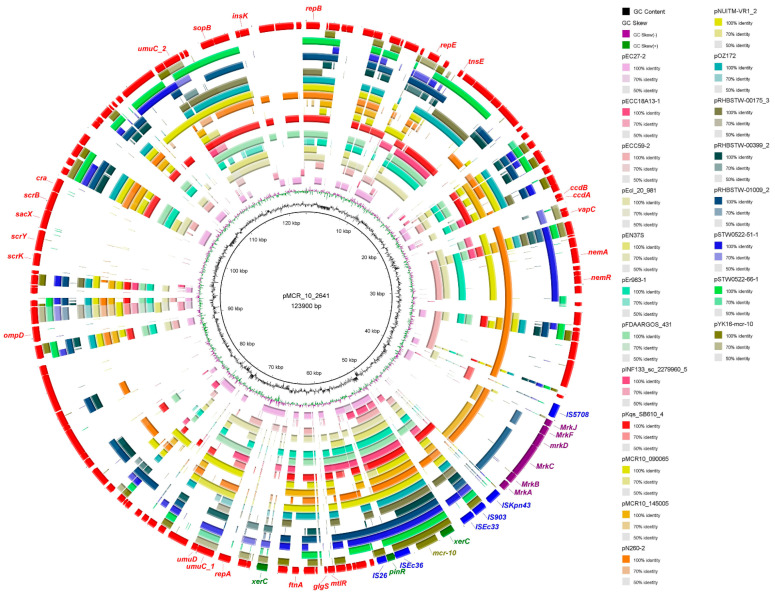
DNA alignment of *mcr-10*-containing plasmids. The out-layer (red color) represents the plasmid pMCR_10_2641 in this study, which was used as the reference plasmid for sequence comparison. Green arrows in the outer circle indicate integrons, blue arrows indicate various insertion sequences, and purple arrows indicate complete virulence islands. Colistin resistance gene *mcr-10* is highlighted by the olive. The arrow direction indicates the direction of transcription. The graph was generated by BRIG v0.95.

**Figure 4 antibiotics-11-00883-f004:**
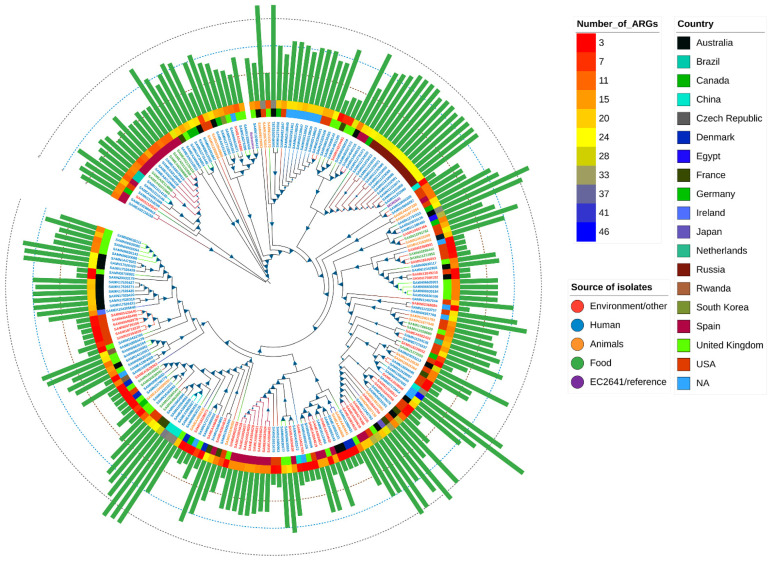
Global phylogenetic analysis of *E. coli* carrying the *mcr* variant. The isolate sequences were compared to the *E. coli* reference genome ATCC25922 by snippy and core SNP comparisons were generated. Maximum likelihood phylogenetic trees were constructed using RAxML-NG. The size of the blue triangle represents the size of the bootstrap value (ranging from 0–100). The label color indicates the source of isolation of the strain; the first circle from the inside is the country information of the isolate, and the second circle is the information of the number abundance of ARGs.

**Table 1 antibiotics-11-00883-t001:** MIC profile of multidrug-resistant *E. coli* isolates.

Isolate ID	MIC (µg/mL)										
	TGC	MEM	AMP	CST	PME	FOX	CRO	CHL	TET	BAC	CPC
EC740	**2**	0.06	**≥256**	1	1	2	<0.6	16	8	**16**	**8**
EC799	**2**	0.03	**32**	2	1	2	<0.6	**32**	**32**	**16**	**8**
EC875	**2**	0.03	**128**	1	1	8	<0.6	**256**	**256**	**16**	**8**
EC2299	**2**	0.06	**≥256**	1	0.5	4	**≥32**	8	**128**	**16**	**8**
EC2639	**2**	0.03	**256**	**8**	**4**	4	<0.6	8	**16**	**16**	**8**
EC2641	**2**	0.12	**32**	**4**	**4**	4	<0.6	16	**16**	**16**	**8**
EC2783	**2**	0.06	**≥256**	0.5	0.5	4	**≥32**	16	**128**	**16**	**8**
SY3705	**2**	0.03	**32**	2	2	16	<0.6	8	4	**16**	**8**

Abbreviation: CRO, ceftriaxone; MEM, meropenem; AMP, ampicillin; FOX, cefoxitin; CHL, chloramphenicol; PMB, polymyxin B; CST, colistin; TGC, tigecycline; TET, tetracycline; BAC, benzalkonium chloride; CPC, cetylpyridinium chloride. The data in bold represent resistance.

**Table 2 antibiotics-11-00883-t002:** Characteristics of quaternary ammonium disinfectant and antibiotic multidrug-resistant *E. coli* isolated from disinfected tableware samples.

Isolate	Plasmid Replicons	Serotype	STs	Antimicrobial Resistance Profile	Resistance Determinants Identified Based on WGS
EC740	Col156	O104: H27	ST2795	TGC-AMP-BAC-CPC	*mdf* (A), *bla*_ZEG-1_, *form**A*
EC799	Col440I, IncFIA (HI1), IncFIB (K), IncY	O81: H9	ST1571	TGC-AMP-CHL-TET-BAC-CPC	*mdf* (A), *bla*_ZEG-1_, *form**A*
EC875	IncFIB (K)	OND: H16	ST4537	TGC-AMP-CHL-TET-BAC-CPC	*aad**A1*, *mdf* (A), *bla*_OXA-10_, *qnr**S1*, *dfr**A14*, *cml**A1*, *flo**R*, *ARR**-2*, *tet* (A), *form**A*, *bla*_MIR-2_
EC2299	IncY	O159: H34	ST218	TGC-AMP-CRO-TET-BAC-CPC	*mdf* (A), *bla*_ZEG-1_, *form**A*
EC2639	Col440I, Col156, IncR	O155: H27	ST2795	TGC-AMP-CST-PME-TET-BAC-CPC	*mdf* (A), *bla*_ZEG-1_, *formA*
EC2641	Col440I, FIA (pBK30683), IncFIB (K), IncHI1A, IncHI1B (R27)	O112ab: H9	ST3907	TGC-AMP-CST-PME-TET-BAC-CPC	*mdf* (A), *mcr**-10*, *bla*_ZEG-1_, *form**A*
EC2783	IncFIA, IncFIB (AP001918), IncFIC (FII), IncI1-I (Alpha), IncX9	O128ac: H34	ST3076	TGC-AMP-CST-PME-TET-BAC-CPC	*bla*_CTX-M-55_, *bla*_TEM-1B_, *qnr**S1*, *tet* (A), *sit*_ABCD_, *strB*, *dfr**A14*, *mdf* (A), *sul**2*
SY3705	IncFIB (K), IncFII, IncY	O83: H19	ST5783	TGC-AMP-BAC-CPC	*mdf* (A), *bla*_ZEG-1_, *formA*

Abbreviation: CRO, ceftriaxone; AMP, ampicillin; CHL, chloramphenicol; PMB, polymyxin B; CST, colistin; TGC, tigecycline; TET, tetracycline; BAC, benzalkonium chloride; CPC, cetylpyridinium chloride. OND, O-antigen not detected.

## Data Availability

The sequencing data generated from this study have been uploaded to the NCBI GenBank database, as detailed in the Materials and Methods section.

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
