# Peer review of "Identification of Mobile Colistin Resistance Gene mcr-10 in Disinfectant and Antibiotic Resistant Escherichia coli from Disinfected Tableware"

_antibiotics, 2022, doi:10.3390/antibiotics11070883_

Round 1

Reviewer 1 Report

The authors presented a manuscript entitled "Identification of mobile colistin resistance gene mcr-10 in disinfectant and antibiotic resistant Escherichia coli from disinfected tableware"; they isolated 161 coliforms from 311 samples of tableware, 8 of them wer identified as E. coli by biochemical tests.

I just have some suggestions:

line 123 Please highlight the importance of the plasmid detected

line 124 Why this analysis was not implemented to the other strains? Or it was made, why is not reported? It si not relevant? please clarify

Author Response

Thanks very much for your detailed comments on our manuscript “Identification of mobile colistin resistance gene mcr-10 in disinfectant and antibiotic resistant Escherichia coli from disinfected tableware” (antibiotics-1764438), which are highly valuable for revising and improving our manuscript. We have studied the comments carefully and made appropriate correction, and hope the revision meet the requirement of Antibiotics. In the revision, we use the "Track Changes" function to make changes to the manuscript. note: The line numbers described in the point-to-point responses are the corresponding line numbers for the simple markers mode of the "Track Changes" function in the revised manuscript. And an item by item response is as follows:

The authors presented a manuscript entitled "Identification of mobile colistin resistance gene mcr-10 in disinfectant and antibiotic resistant Escherichia coli from disinfected tableware"; they isolated 161 coliforms from 311 samples of tableware, 8 of them were identified as E. coli by biochemical tests. I just have some suggestions:

Point 1: line 123 Please highlight the importance of the plasmid detected.

Response 1: Thank you very much for your kind suggestions. The mcr-10-bearing IncFIB-type plasmid pMCR_10_2641 is an antimicrobial resistance and virulence co-existing plasmid, which may have important implications for clinical treatment. In fact, IncFIB-type plasmids usually carry multiple resistance genes, and a recent study reported a novel resistance gene cluster tnfxB1-tmexCD1-toprJ1 located in an IncFIB-type plasmid carrying multiple resistance genes (strAB, armA, aph(3')- ia, qnrB4, sul1, mphA, mphE, msrE, and bladha1) (https://doi.org/10.1128/AAC.02340-20). We have modified this part to give pertinent description in the discussion section of revision (lines 291-296).

Point 2: 124 Why this analysis was not implemented to the other strains? Or it was made, why is not reported? It is not relevant? please clarify.

Response 2: Thank your reminding. It would be more appropriate to perform complete sequence analysis of plasmids for all strains by second and third generation whole genome sequencing techniques. For some practical reasons all of our strains were first screened by second generation whole-genome sequencing to obtain strains for which we needed further third generation whole genome sequencing to resolve the complete sequences. Nevertheless, we have fully accepted your suggestion to add the plasmid replicon results from the second generation sequencing analysis to the results section 2.2 of revision (lines 115-117).

Reviewer 2 Report

Dear authors, I read with a high interest your paper on ''Identification of mobile colistin resistance gene mcr-10 in disinfectant and antibiotic-resistant Escherichia coli from disinfected tableware''. However, some concerns should be addressed to improve the quality of the manuscript.

Areas of concern:

Introduction

Line 40: antimicrobial instead of bactericidal since its broad-spectrum activity comprises activities against bacteria, algae, fungi, and viruses.

Lines 56-57: the following phrase is not grammatically correct, a word seems to be missing: a cationic polypeptide antibiotic that was one of the first antibiotics significantly against Gram-negative bacteria.

Lines 76-77: The authors should justify their choice on E.coli among foodborne bacteria. Is it due to a high prevalence of mobile colistin resistance genes among E.coli strains in general or due to their importance in foodborne diseases in China?

Results

Line 100: insert ‘’to’’ between ‘’resistance’’ and ‘’including’’

Line 111: remove ‘’s’’ from ‘’disinfectants’’

Lines 126-127: The authors should give the name of the uropathogenic specific protein mentioned in this study

Line 132: insert ‘’a’’ before ‘’4-fold’’

Line 134: insert ‘’a’’ before ‘’sequence’’

Lines 138-147: The authors seem to be discussing their results while there is a dedicated section for the discussions

Line 152: insert ‘’a’’ before ‘’sequence’’

Lines 175-178: The authors seem to be discussing their results while there is a dedicated section for the discussions

Line 215: put ‘’a comma’’ after ‘’benzalkonium chloride’’

Line 216: remove ‘’a comma’’ after ‘’cetylpyridinium chloride’’

Line 225: add ‘’s’’ to ‘’human’’ and put ‘’a comma’’ after ‘’environment’’

Discussion

Lines 243: remove ‘’of’’ before ‘’resort’’

Lines 211: insert ‘’a comma’’ after ‘’ imipenem’’

Lines 317-318: insert ‘’a comma’’ after ‘’gentamicin’’

Line 321: add ‘’s’’ to ‘’ human’’

Materials and methods

Line 337: 311- Never start a sentence with a number

Line 338: How did you collect the samples from the tableware? Using wet swabs?

Line 345: Can you justify the choice of the following parameters: 2000 μg/mL for 30 min at room temperature?

Line 85: use µg instead of ug (verify this across the entire manuscript).

Lines 396-398: use past tense to describe the procedures contained in this section

Lines 400-413: specify the algorithms used for the different bio-informatics analyses

Conclusion

Line 434: add ‘’s’’ to ‘’ isolate’’

References

Line 475 (reference 5): check for consistency

References 9, 13, 23, 27, and 28: which format authorizes you to cite 10 authors before using et al? Check if you should cite six or ten authors before you add et al.

Author Response

Thanks very much for your detailed comments on our manuscript “Identification of mobile colistin resistance gene mcr-10 in disinfectant and antibiotic resistant Escherichia coli from disinfected tableware” (antibiotics-1764438), which are highly valuable for revising and improving our manuscript. We have studied the comments carefully and made appropriate correction, and hope the revision meet the requirement of Antibiotics. In the revision, we use the "Track Changes" function to make changes to the manuscript. note: The line numbers described in the point-to-point responses are the corresponding line numbers for the simple markers mode of the "Track Changes" function in the revised manuscript. And an item by item response is as follows:

Dear authors, I read with a high interest your paper on ''Identification of mobile colistin resistance gene mcr-10 in disinfectant and antibiotic-resistant Escherichia coli from disinfected tableware''. However, some concerns should be addressed to improve the quality of the manuscript.

Areas of concern:

Introduction

Point 1: Line 40: antimicrobial instead of bactericidal since its broad-spectrum activity comprises activities against bacteria, algae, fungi, and viruses.

Response 1: Thank your careful correction, and we have revised it (line 43).

Point 2: Lines 56-57: the following phrase is not grammatically correct, a word seems to be missing: a cationic polypeptide antibiotic that was one of the first antibiotics significantly against Gram-negative bacteria.

Response 2: Thanks for your corrections on grammar and spelling issues in our manuscript. We have made the modification according to your suggestion (lines 58-60).

Point 3: Lines 76-77: The authors should justify their choice on E.coli among foodborne bacteria. Is it due to a high prevalence of mobile colistin resistance genes among E.coli strains in general or due to their importance in foodborne diseases in China?

Response 3: Thanks for your professional advice. E. coli, as an important opportunistic pathogen among foodborne bacteria, is one of the common outbreak factors of foodborne diseases in China. Monitoring bacterial multidrug resistance to disinfectants and antibiotics in E. coli from disinfected tableware in the catering industry is important to understand the spread of resistance genes in foodborne bacteria and to regulate sanitization practices. In addition, according to the proportion of host distribution of mobile colistin resistance genes reported by NCBI, E. coli accounted for the largest proportion (49.7%). These are the reasons why we chose E. coli from food-borne bacteria as the subject of our study, and we made changes on the corresponding parts of the revised manuscript (lines 76-80).

Results

Point 4: Line 100: insert ‘’to’’ between ‘’resistance’’ and ‘’including’’.

Response 4: Done accordingly (line 106).

Point 5: Line 111: remove ‘’s’’ from ‘’disinfectants’’.

Response 5: We have modified at the corresponding (line 119) and make consistent changes to the revised version.

Point 6: Lines 126-127: The authors should give the name of the uropathogenic specific protein mentioned in this study

Response 6: The information has been added in the revision (line 135).

Point 7: Line 132: insert ‘’a’’ before ‘’4-fold’’.

Response 7: Done accordingly (line 140).

Point 8: Line 134: insert ‘’a’’ before ‘’sequence’’.

Response 8: Double checked and revised. Thanks for this reminding (line 142).

Point 9: Lines 138-147: The authors seem to be discussing their results while there is a dedicated section for the discussions.

Response 9: We have revised this by relocating part of the content in the discussion section (lines 271-276).

Point 10: Line 152: insert ‘’a’’ before ‘’sequence’’.

Response 10: Done accordingly (line 156).

Point 11: Lines 175-178: The authors seem to be discussing their results while there is a dedicated section for the discussions.

Response 11: We have reorganized this section into a discussion section (lines 300-304).

Point 12: Line 215: put ‘’a comma’’ after ‘’benzalkonium chloride’’.

Response 12: Done accordingly (line 214). We have corrected similar issues throughout the text, and thank you again for your thorough and careful review of our manuscript.

Point 13: Line 216: remove ‘’a comma’’ after ‘’cetylpyridinium chloride’’.

Response 13: Done accordingly (line 215).

Point 14: Line 225: add ‘’s’’ to ‘’human’’ and put ‘’a comma’’ after ‘’environment’’.

Response 14: Done accordingly (line 224).

Discussion

Point 15: Lines 243: remove ‘’of’’ before ‘’resort’’.

Response 15: Done accordingly (line 242).

Point 16: Lines 311: insert ‘’a comma’’ after ‘’imipenem’’.

Response 16: Done accordingly (line 324).

Point 17: Lines 317-318: insert ‘’a comma’’ after ‘’gentamicin’’.

Response 17: Done accordingly (line 331).

Point 18: Line 321: add ‘’s’’ to ‘’ human’’.

Response 18: Done accordingly (line 335).

Materials and methods

Point 19: Line 337: 311- Never start a sentence with a number.

Response 19: Done accordingly (lines 350-352).

Point 20: Line 338: How did you collect the samples from the tableware? Using wet swabs?

Response 20: First of all, thank you for your hard work on correcting our manuscript. In China, the microbiological sampling method for tableware refers to paper method by Chinese National Standard GB 14934-2016. Taking chopsticks as an example, five chopsticks were used as one sample, and the fast test paper for coliform of tableware (NANJING SAN-AI, China ) were wet with sterile phosphate buffered saline and immediately applied to the inlet end of the chopsticks. Each sample was coated with two sheets of rapid test paper and put into sterile bags. For other food and beverage devices, stick the moistened test sheets on the contact surface of food and mouth for 30 seconds and then take it out and put it into sterile bags. Then the coliform detection sheets were placed in a 37℃ incubator for overnight culture. When the paper turned yellow or showed red spots on a yellow background, it was judged to be positive for coli-form. Coliform were collected from positive sheets with wet cotton swabs for subsequent microbiological analysis. We have made modifications in the Materials and Methods part of the revision (lines 350-358).

Point 21: Line 345: Can you justify the choice of the following parameters: 2000 μg/mL for 30 min at room temperature?

Response 21: Considering the actual Chinese standard of disinfection behavior, we referred to the Chinese National Standard GB/T 26369-2010. The positive sample of fast test paper for coliform of tableware was regarded as the tableware contaminated with coliform. According to the standard, the contaminated samples were soaked for 5 min to 20 min with the disinfection concentration of 400 ~ 12000 μg/mL quaternary ammonium. Combined with our previous experimental experience, the disinfection condition of 2000 μg/mL for 30 min at room temperature was beneficial to the separation of the resistant bacteria of quaternary ammonium disinfectant.

Point 22: Line 85: use µg instead of ug (verify this across the entire manuscript).

Response 22: The error has been modified in the full text (line 90).

Point 23: Lines 396-398: use past tense to describe the procedures contained in this section.

Response 23: Done accordingly (line 414-416).

Point 24: Lines 400-413: specify the algorithms used for the different bio-informatics analyses

Response 24: Thank you for your suggestion, we have specified the algorithms and parameters used for the corresponding bioinformatics analysis tools. And if not specified otherwise, the default parameters are used. The relevant section has been revised (417-436).

Conclusion

Point 25: Line 434: add ‘’s’’ to ‘’ isolate’’

Response 25: Done accordingly (line 457).

References

Point 26: Line 475 (reference 5): check for consistency

Response 26: Thanks to the academic rigor of the reviewers, and we have made a thorough checking of the consistency of the cited literature and revised accordingly (line 45).

Point 27: References 9, 13, 23, 27, and 28: which format authorizes you to cite 10 authors before using et al? Check if you should cite six or ten authors before you add et al.

Response 27: Thank you for your valuable comments on the reference format, which we take very seriously. We carefully examined Reference List and Citations Style Guide for MDPI Journals (https://www.mdpi.com/authors/references): For documents with a large number of coauthors (more than 10), we should either cite all authors, or cite the first ten authors, then add a semicolon and add ‘et al.’. Here we use the latter approach to uniformly specify the citation format for references.

Reviewer 3 Report

The work by Zhang et al seem to be interesting as well as laborious. I have some reservations before accepting the manuscript the supplementary file is corrupt. I cannot see it. The data from the PCR is nowhere to be found. There is not statistics on the MIC value determination. The supplementary file is corrupt too. I cannot see it. Please add the data in the main manuscript so that the review process can be redone

Author Response

Thanks very much for your detailed comments on our manuscript “Identification of mobile colistin resistance gene mcr-10 in disinfectant and antibiotic resistant Escherichia coli from disinfected tableware” (antibiotics-1764438), which are highly valuable for revising and improving our manuscript. We have studied the comments carefully and made appropriate correction, and hope the revision meet the requirement of Antibiotics. In the revision, we use the "Track Changes" function to make changes to the manuscript. note: The line numbers described in the point-to-point responses are the corresponding line numbers for the simple markers mode of the "Track Changes" function in the revised manuscript. And an item by item response is as follows:

The work by Zhang et al seem to be interesting as well as laborious. I have some reservations before accepting the manuscript the supplementary file is corrupt. I cannot see it. The data from the PCR is nowhere to be found. There is not statistics on the MIC value determination. The supplementary file is corrupt too. I cannot see it. Please add the data in the main manuscript so that the review process can be redone.

Response: We apologize that you were unable to view the supplemental data due to an error when we compressed the supplemental material file. We have re-added the PCR data to the supplementary material, and even put all the supplementary material to the end of the revised version (lines 631-651), considering the convenience for the reviewers. And the MIC values have been added to lines 140-141 of the revision. Thank you again for your professional comments.

Round 2

Reviewer 2 Report

Materials and Methods

Line 367: Replace ''were'' with ''is''  fast test paper is singular.

Reviewer 3 Report

The manuscript looks much improved. 

Can be accepted in the present form.